# Compatibility of Small Team Personalities in Computer-Based Tasks

Angeliki Antoniou

Department of Informatics and Telecommunications, University of Peloponnese, 22100 Tripolis, Greece;
angelant@uop.gr

**Abstract:** There are works that study personality and task performance but there are no (or very few works) that study the balancing of personalities within teams that work together towards a common goal in computer-based tasks. This study investigates how personality compatibility in collaborative tasks affects performance, intra-group communication and participants' emotions for computer-based tasks and introduces the challenges for research in this field. Using the DISC (dominance, inducement/influence, submission/steadiness, compliance) tool for personality assessment and team compatibility, 12 teams were created with either balanced or imbalanced personality compositions. Results showed statistical differences in emotions between the two experimental conditions and also differences in terms of time needed for the completion of the game. The present work showed the qualitative differences between cooperative tasks and revealed the challenges of studying further team compatibility for different tasks.

**Keywords:** personality-based balancing; team formation; team compatibility

## 1. Introduction

Previous research has found that compatibility of the personalities of individuals within teams is very important for the collaboration of those individuals, affecting work performance [1]. Psychology is involved in the study of how people's personalities affect the way they process information, reach conclusions and act, as well as the way they function within teams [2]. Good group dynamics imply a positive environment and increased work efficiency [3]. However, when team members do not have compatible personalities then negative phenomena are observed like tension, conflict and stress [4,5].

Previous works study team members' compatibility in a variety of environments and tasks, like face-to-face interaction [1,3,6–9], computer-based communication [6,10,11], etc. Cooperative tasks, whether these are collaborative or competitive, require the effective communication, coordination and function of teams [12,13]. Due to its importance, there are numerous works that study intra-group cooperation and they identify different important factors for effective cooperation like group size [14–18], moral preferences [19] and group members' cooperation motives [20,21]. There are fewer works that focus on individual group member's personality aspects and why these personalities are compatible or not with the other group members [11].

In addition, for computer-based tasks when virtual teams and face-to-face ones were compared, the complexity of the endeavor became apparent, since many factors seem to play an important role in the effectiveness of a team [22]. It seems that compared to face-to-face teams, virtual ones need good supporting tools in order to perform their tasks and to increase group cohesiveness [23]. Face-to-face teams seem to outperform virtual ones under certain circumstances. For example, face-to-face teams are better for decision-making tasks and for tasks with a single right answer [24], they also report higher levels of satisfaction, although their communication effectiveness seems to be similar to the virtual teams [25]. The complexity of the field requires further study.

Thus, this paper will examine the way personality-balanced and -imbalanced teams function in terms of time efficiency, task effectiveness, communication practices and emotions experienced by the team members all within the framework of short cooperative tasks. We present the way balanced and imbalanced teams were created and analyze the outcomes of their interactions as well as their reports on communication quality and the emotions participants felt. The results provide implications for how to use personality testing to form effective teams within the context of cooperative tasks. The work shows the challenge in studying group compatibility for different computer-supported tasks and reveals its complex nature. The compatibility of teams in different computer-supported tasks is a relatively new field that requires further study, due to the significant implications it might have in increasing productivity, engagement and satisfaction in team work. Being affected by multiple factors, like nature of the task, level of computer support required, cultural aspects, etc., group processes like those described here, raise new challenges in research due to the multidisciplinary nature of the field, involving social psychology, human factors and technology.

Finally, the work is structured in a way that reflects real-life work situations (e.g., free communication of participants, participants known to each other, familiar space, etc) but it also builds on the assumption that computer-based tasks differ qualitatively from tasks that do not require the use of a computer, since technology changes the very nature of work and does not simply function as yet another office tool [26].

## 2. Literature Review

It is known that personality significantly affects work performance, since different individuals have different perceptions of their work environment and interact differently with it [27]. Personality not only affects how people interact with their work object, but also with each other within their teams. In fact, face-to-face interaction of people with different personality traits within teams has been well studied [7]. Team-building processes are also well reviewed [3] and it seems that when teams are balanced in terms of complementary personalities of their members they produce better results [9]. It is also known that effective teams have leaders of different types, since there are more than one leader-type personalities, and the existence of similar leader types is avoided [28]. The importance of personality matching in teams is studied in different settings, like face-to-face interactions [8,29], computer-based communication [30], virtual environments [6], gaming engagement [31], gaming decision-making [32] and crowdsourcing platforms [11]. Moreover, in regards to job performance, personality is crucial since not only the quality of the outcome is determined but even whether the task will be completed or not [33].

In addition, in order to build balanced teams, an appropriate tool is needed. There are numerous tools that assess individual personality traits, such as Costa and McCrae's [34] NEO-PI-R five factor analysis, Holland's 6 personality types [35], or Eysenck's supertraits [36] but these primarily study the person as an individual. There are other tools that study the group dynamics as a whole (e.g., collective intelligence, group performance, etc) [37,38]. However, none of the above tools is suitable since the present work wishes to see how the individual's personality emerges within a team and how it affects the roles that the different team members assume. For this purpose, a tool that assesses the individual and the role she will have inside a team was needed. After reviewing the available tools for such a purpose [3], the DISC (dominance, inducement/influence, submission/steadiness, compliance) tool [39] was chosen for 4 reasons: 1) DISC is more flexible than other tools since it requires a rather small amount of people per team (only four), 2) DISC has been successfully used recently to assess the personalities of members of collaborating teams and study team performance [11,40], 3) DISC is widely used in human resources management due to its practicality and tangible outcomes [40,41], 4) DISC is a reliable and valid tool based on recent studies [42]. The latter reason is of importance since the present work aims at applying research outcomes to real work settings and thus targets actual organizational issues. Thus, a tool that has been successfully used in real work settings is appropriate here. DISC is based on temperament theory and focuses on relatively stable personality

characteristics, like extraversion/introversion [43] and is dedicated to the study of the way people interact inside a team based on their individual personalities and the roles they wish to have (e.g., leader, non-leader). DISC identifies four main types of personalities inside teams, the task-oriented leaders (D leaders), the socio-emotional leaders (I leaders), the socio-emotional non-leaders (S non-leaders) and the task-oriented non-leaders (C non leaders) (Figure 1). According to DISC, balanced teams should have one of each type.

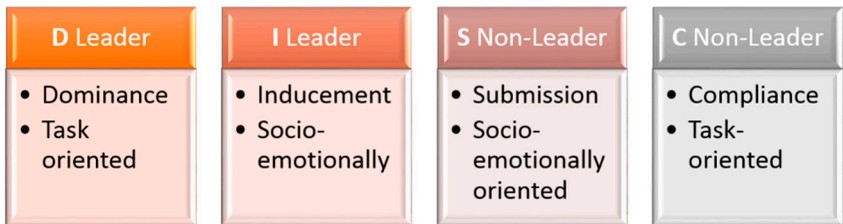

**Figure 1.** DISC (dominance, inducement/influence, submission/steadiness, compliance tool) personality types (adapted from [11]).

Moreover, the literature emphasizes the importance of the task's nature, i.e. that a person's work behavior is highly related to the task that the person is involved in. Thus, it is important to separate tasks and have a task typology, since different personality characteristics seem to have bidirectional relationships with different task types and dimensions of job performance [44]. In addition, task type is found to influence effectiveness significantly and cooperative tasks especially were found to have better results in terms of achievement and productivity from other type of tasks like interpersonal competition or simple individual effort [45]. The preference for cooperation also seems to be personality related, since not all personality types like cooperative tasks [46]. The effort made by different group members seems to be a combination between their personalities and the nature of the task, implying that certain personalities will give more effort in specific tasks of different characteristics [47].

Thus, the nature of the different tasks had to be studied, too. After reviewing the Steiner task typology [48] and other available literature [49], we can identify 7 main task types (Table 1). Task type I denotes whether a task can be divided to further subtasks or not (values: 1. divisible – existence of subtasks, 2. unitary – no substasks). Task type II denotes whether the team focuses on the quality or the quantity of the task (values: 1. maximizing – importance placed on quantity, 2. optimizing – importance placed on quality). Task type III denotes the mechanism used by the team to combine the contributions of its individual members (values: 1. additive – individual inputs are added, 2. compensatory – group product is the average of individual judgments, 3. disjunctive – product is selected from pool of individual judgments, 4. conjunctive – product is a synthesis of all member contributions, 5. discretionary – group can decide how individual inputs relate to group product). Task type IV denotes the way of co-operation among the team members (values: 1. collaborative – commonality of interests, 2. competitive – conflict of interests, 3. mixed motive – both common and conflicting interests). Task type V denotes the level of difficulty of the task (values: 1. easy or 2. difficult tasks). Task type VI denotes the task's duration (values: 1. Short or 2. long tasks). Task type VII denotes the subjectivity of the task (values: 1. Intellective – there is a correct answer, 2. judgmental - no demonstrably correct answer).

In a previous work [11], the above typology was used and team formation was tested for a divisible, optimizing, discretionary, collaborative, judgmental and medium duration task during which the team members had to create a product commercial. In that study [11], the participants had to collaborate on a Google document in order to produce a product advertisement, without being able to communicate with each other synchronously but only through comments on the document. Also working through a crowdsourcing platform, the participants did not know each other and were of different cultural backgrounds. The task was of medium duration, since participants had a week in

order to complete the task. The results showed that balanced teams produced significantly better products and reported more positive emotions.

**Table 1.** Current task in the task typology.

| Task Type | Description | Values |
|---|---|---|
| I | Whether a task can be divided to further subtasks or not | 1.Divisible—existence of subtasks<br>2.Unitary—no subtasks |
| II | Whether the team focuses on the quality or the quantity of the task | 1. Maximizing—importance placed on quantity<br>2. Optimizing—importance placed on quality |
| III | Mechanism used by the team to combine the contributions of its individual members | 1. Additive—individual inputs are added<br>2. Compensatory—group product is the average of individual judgments<br>3. Disjunctive—product is selected from pool of individual judgments<br>4. Conjunctive—product is a synthesis of all member contributions<br>5. Discretionary—group can decide how individual inputs relate to group product |
| IV | Way of co-operation among the team members | 1. Collaborative—commonality of interests<br>2. Competitive—conflict of interests<br>3. Mixed motive—both common and conflicting interests |
| V | Level of difficulty of the task | 1. Easy<br>2. Difficult |
| VI | Task's duration | 1. Short<br>2. Medium<br>3. Long tasks |
| VII | Subjectivity of the task | 1. Intellective—there is a correct answer<br>2. Judgmental—no demonstrably correct answer |

In the present work, the same typology and type of task was used, but this time the task was divisible, optimizing, discretional, collaborative, intellective and short. In addition, the task required face-to-face interaction of the group members that knew each other from before, as opposed to [11] that used a computer-based communication for two main reasons: 1) the communication medium seems to be very important, affecting the levels of member interdependence, as well as the amount and the quality of the communication [50] and 2) the present task imitates real work environments that often people who know each other can discuss and decide face-to-face, while using computers to complete their work.

Therefore, the importance of the task's nature is investigated in the present work, studying their possible qualitative differences and further building on the previous work by [11], in an attempt to complete the study of different tasks and the combination of different factors in light of personality compatibility. Being a relatively new field, one that combines group compatibility in different computer supported tasks, the present work also introduces the challenges of the study. Thus, from all the above the present work's research hypotheses are formed as follows:

**H1:** *Balanced teams will be more effective in finding the task solution than imbalanced ones.*

**H2:** *Balanced teams will report more positive achievement emotions compared to imbalanced teams.*

**H3:** *Balanced teams will report better communication between the team members compared to imbalanced teams.*

**H4:** *Balanced and imbalanced teams will need different time duration to complete the task.*

## 3. Research Methodology

### 3.1. Participants

A between subjects design was followed in order to collect research data. All participants were undergraduate students of the Department of Informatics and Telecommunications (University of Peloponnese, Greece), and the total number of participants was N = 50 (15 females, 35 males). There were 7 imbalanced teams and 5 balanced. Initially there were 6 teams in each experimental condition, but one group changed half way from balanced to imbalanced, since the D leader left. All teams had 4 or 5 members (only one team had 3 members because the 4th member did not show

up). The following table (Table 2) shows the distribution of male and female participants in the two experimental conditions.

**Table 2.** Participants.

|  | Imbalanced Teams | Balanced Teams |
|---|---|---|
| Females | 10 | 5 |
| Males | 19 | 16 |

### 3.2. Team Formation

Once the participants arrived they were asked to complete a questionnaire to identify their DISC type. In a compatible team, all DISC types were present. If there were five people in a balanced team the remaining individual was a non-leader type, either S or C. In imbalanced teams the DISC balance was not followed and, for example, there were teams with 3 D leaders or no leader type at all. Figure 2 describes the team formation processes.

| Balanced Teams | Imbalanced Teams |
|---|---|
| D + I + S + C<br>+<br>S or C | D + D<br>+<br>D or C + C + C<br>+<br>D or S |
| **Imbalanced Team without D leader** | **Imbalanced Leaderless Team** |
| I + S + C | C + C + C + C |

**Figure 2.** Team formation processes based on the DISC tool.

### 3.3. The Task

After the teams were formed, they were all given a page with instructions. The teams had to access a folder and see three images. They were asked to write the similarities between these images of three women. They were also given a tip: "You need to know who they are". The time that each team needed to complete the task was also recorded and the teams were informed about time measurement, as well.

The task was simple, but they had to collaborate and think as a group in order to find the solution. All teams accessed the same folder with the images of the three women shown in Figure 3. In order to find the solution, the teams had to think to perform image search (not all teams found the solution).

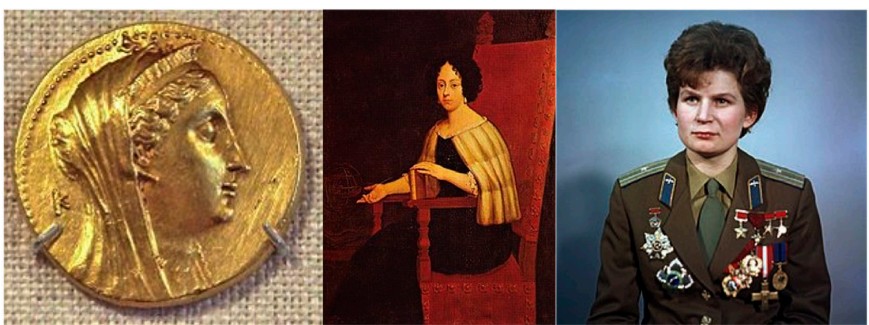

**Figure 3.** Task images [task solution: the main element that connects the three women is the fact that they were the first to achieve something. The woman in the coin is Arsinoe II, the first mortal female depicted on a coin (Egypt, 3rd century BC). The woman in the middle is Elena Cornaro Piscopia, the first woman to receive a University degree (University of Padua, 1678). The woman on the right is Valentina Tereshkova, the first woman in space (USSR, 1963)].

The task used is similar to real work settings where collaboration is needed together with the use of information technology (IT). In this case, the task requires people to access files of digital images, perform digital image search, access relevant informative material, and organize findings while discussing and deciding on the use all the resources available to them.

### 3.4. After the Task

After the teams decided that they had found all the similarities, individual questionnaires were distributed to measure self-reported emotions (Pekrun achievement emotions questionnaire—[51]) and to capture team communication aspects (4 questions used asking for: 1. satisfaction about team's performance, 2. satisfaction on the quality of team communication, 3. ease of sharing personal opinions, 4. perceptions of appreciation and acceptance by the other team members).

## 4. Results

The results were analyzed on two levels. The first level was related to the research hypotheses and it studied the performance, emotions and communications of the teams. A Mann–Whitney U test was performed in order to test the four hypotheses. This non-parametric test was used, since the sample did not follow the normal distribution. On the second level, we looked deeper to study the individual communication and emotions of the people inside the teams. For this analysis, chi square tests were performed.

### 4.1. Team Level

H1: Once balanced and imbalanced teams were compared in regards to the task solution, no statistical significance was found. In particular, balanced teams found the solution 2 out of 5 times and the imbalanced teams found the solution 4 out of 7 times.

H2: Significant differences were found between the two experimental conditions for the self-reported feeling of boredom. The Mann–Whitney U test value was $U = 230$, $p = 0.015$. However, the results were exactly opposite from the ones expected and it was the imbalanced team members that reported the lowest boredom levels.

H3: The statistical analysis did not reveal any significant differences between the two experimental conditions in regards to the quality of communication among the team members.

H4: The balanced teams needed significantly more time than the imbalanced teams to complete the task. The Mann–Whitney U test value was $U = 142$, $p = 0.000$. Again, the results were opposite from the expected ones, since time efficiency was expected from the balanced teams; however, the contrary was found.

### 4.2. Individual Level

In addition to the Mann–Whitney U test that studied the main research hypotheses, chi square tests were also performed in order to study in depth what happened inside the teams. In particular, the emotions of different DISC type members were analyzed, as well as perception about their quality of communication. It was found that S types felt more embarrassed than other DISC types, $\chi^2(3) = 7.729$, $p = 0.050$. It was also found that different DISC types experienced different levels of boredom, since $\chi^2(6) = 19.485$, $p = 0.003$. The D leaders reported that they were not bored, whereas C non-leaders reported the highest boredom levels. Finally, leader types (D,I) significantly more times reported that the team accepted their opinion compared to non leader types (S,C) and $\chi^2(6) = 15.044$, $p = 0.020$.

## 5. Findings and Discussion

The discussion section also follows the two levels of analysis as they were presented in the results section.

### 5.1. Team Level

H1: The first hypothesis was not supported by the current research results (failed to reject the null hypothesis: "There will be no difference in two experimental conditions in regards to reaching the task solution"). In the described experimental conditions the balanced teams did not differ from the imbalanced ones when it came to finding the solution of the problem. Compared to [11] where very significant differences were observed in the outcome of the teams' works, the current results might be due to two reasons. The nature of the task itself, and the fact that the team members participating in the task knew each other, might have affected the performance. As in real work settings where people know each other in various degrees, the participants of the current work were all undergraduate students from the Department of Informatics and Telecommunications (University of Peloponnese), which is a small department located in a small city. In addition, students are accustomed to group work, since most modules require group course work. The habit of working in groups and the previous contact with each other might have affected the team formation processes and/or probably does not allow the DISC types and the expected group dynamics to emerge as it would do so if participants do not know each other. These differences in task performance between groups that know each other and groups that do not need to be further studied, since they can significantly change the way work groups are formed for the different tasks.

H2: When the self reported emotions of the two experimental conditions were compared, the members of the imbalanced teams were found to be less bored than the members of the balanced teams. The imbalanced teams' group dynamics seemed to reduce boredom, meaning that task engagement was probably increased. Thus, the potential of imbalanced teams can be further investigated in cooperative tasks when the goal is boredom reduction.

H3: The third hypothesis was not confirmed, meaning that no differences in communication quality were reported in the two experimental conditions. Again, compared to [11] differences in the quality of communication were reported, the current findings might also be due to the fact that the participants knew each other and were used to working together.

H4: The fourth hypothesis was confirmed by the results (the null hypothesis was rejected: "There will be no differences in the time needed for the completion of the task for the two experimental conditions"). The teams in the two experimental conditions needed different times to complete the task. Balanced teams engaged with the task for more than 10 minutes on average (mean = 14.6 minutes, standard deviation = 8.61), whereas imbalanced teams engaged less than 10 minutes on average (mean = 9.8 minutes, standard deviation = 3.84). Although this was an unexpected finding, based on previous research that had shown the time efficiency of balanced groups, this result has to be viewed in light of the qualitative differences between the different tasks. One hypothesis could be that group members of balanced teams experienced greater enjoyment in communicating with each other that needed more time to reach a conclusion. A balanced team implies more harmonious communication between the team members and this longer engagement with the task might reflect that. When long task duration is a goal of a collaborative activity, as it could be in educational settings, for example, then balanced team formation needs to be considered. However, this is only an assumption that was not investigated in the present work and definitely requires further exploration.

When time spent was also seeing in the light of solving the task problem (only calculated the time from teams that found the task solution), then again the balanced teams that solved the problem needed on average 16.5 minutes whereas the imbalanced teams needed almost half the time (mean = 8.75). However, this finding cannot be discussed further due to the limited number of teams that managed to solve the task (only 6 teams in total), and is discussed here as a possible tendency we observed, requiring further studying.

### 5.2. Individual Level

Very interesting elements from the team's processes and the impact on the individual level came up, once the individual members were studied closer. Significantly more S types, non-leaders, reported

feeling more embarrassed during their interaction with the other team members. Non-leader types also have increased Introversion levels, as it can be seen from the relation between the DISC tool and the MBTI tool (Myers-Briggs Type Indicator) that captures cognitive style [52,53]. The embarrassment reported by S type individuals might be connected to their introversion traits.

In addition, D leaders seemed to be more engaged and reported less bored than non-leader types, like C individuals. Leader types seem to be more talkative, take more initiatives and in general they seem to have higher energy levels. Again according to the correlation between the DISC types and MBTI types, DISC leaders seem to be extroverts. Extrovert leaders inside teams seem to have active roles, which could explain the low boredom levels they experienced. On the other hand, non- leader types like C, reported being more bored than any of the other types, keeping a less active and a quieter role.

Finally, it was not surprising to find that all leader types (both D and I) reported that their opinions were well accepted by their teams, in comparison to non-leader types like C and S that reported lower opinion acceptance levels. By definition, leaders' opinions are well received and respected in teams and these same patterns were also recorded in the present work.

It is also interesting to note how the different teams used the computers to organize their work. After observing the teams, we found out that all of them used only a single computer around which all team members gathered, although computers were available for all team members if they wished to work separately. Only one team member was using the computer (not taking turns with other team members), usually following advice from the other participants. In almost all teams, while one person was using the computer, another one kept some handwritten notes. After the end of their search, the handwritten notes were copied on a Word document and were saved on the computer. This phenomenon might be due to two main reasons: 1) The task was short and teams might have felt that this is the most time efficient manner to work and 2) there might be cultural differences that affect group behavior, since the experiments took place in Greece which is by and large a collective culture (as opposed to more individualistic ones). In that light, team members might have felt the need to gather around one computer, rather than using different devices individually. This interesting finding is worth investigating further and results should be compared with data from teams of different cultural backgrounds, since it is known that culture affects people's behavior within groups [54]. Finally, Table 3 summarizes the results of the current work and compares them with the reference work of [11] in order to provide a quick visualization of results.

**Table 3.** Comparison of current study with reference study [11].

|  | Present work | Reference Study [11] |
| --- | --- | --- |
| Task | Divisible<br>Optimizing<br>Discretionary<br>Collaborative<br>Easy<br>Short<br>Intellective | Divisible<br>Optimizing<br>Discretionary<br>Collaborative<br>Easy<br>Medium<br>Judgmental |
| Method | Face-to-face, members knew each other from before. All participants were Greek, undergraduate students. | Asynchronous through crowdsourcing platform. Members did not know each other from before. Participants from multiple cultural backgrounds. |
| Finding 1 | No difference in the task outcome. Balanced and imbalanced teams equally found the task solution. | Significant differences in the task outcome. Balanced teams produced better product commercials. |
| Finding 2 | When emotions were measured, imbalanced team members reported significantly lower levels of boredom. High acceptance levels were only reported by leader type personalities. | Balanced team members reported significantly higher levels of satisfaction, lower levels of in-group conflict and higher levels of acceptance of members' individual opinions by the team. |
| Finding 3 | No differences in the quality of intra-group communication. | Balanced teams reported increased quality of intra-group communication. |

## 6. Conclusions and Recommendations for Future Work

The significant findings of the present work not only provide an insight into group dynamics during collaborative tasks but also reveal the differences observed in such tasks compared to other tasks, studied previously, introducing the challenges in a relatively new field. For example, important differences from [11] were recorded, although it was a similar type of task (short, conjunctive, cooperative) but not an identical task to the present one (short, discretionary, cooperative). The task type seems to affect group dynamics very differently. Team compatibility and balance is still relevant as in other studies, producing significant results but in a different way, since in this case, opposite processes were observed (e.g., reduced time for imbalanced teams). In this light, it makes sense to have a task's typology to continue with the study of team formation in cooperative tasks (including competitive and collaborative ones). The nature of the task seems to affect group dynamics and it should be studied further. Being one of the first works to study personality compatibility for computer-based short tasks, the present work used a limited sample of participants to study possible tendencies. Since very interesting tendencies were observed and in some case reverse tendencies from the expected ones, the study will be repeated with more participants and groups. In addition, this was one of the first works to study team building for short cooperative tasks. Despite the small sample, the present work showed the importance of studying team-building processes and personality compatibility specifically for different task types. Although the present work is an observational study that does not imply causality (as also reflected in the statistical analysis used), it is important to note that the small sample size possibly affects the reproducability of results [55]. Replication studies are needed before certain conclusions can be drawn during which the present work can function as a pilot study.

In addition, in the present work the participants knew each other from before and all came from a single cultural background. This provides another key difference between this work and the previous ones. Again, this approach of using people from a single background who know each other from before seems close to real work settings. In our future work the experiments will be repeated with strangers in order to further study the possible effects of previous meetings of group members. It is also interesting to add the cultural aspects that might affect the processes described. Although in [11], participants came from all over the world, in the present work all participants were Greek (south Mediterranean collective culture), implying that some of the differences observed might be due to the sampling process. Nevertheless, the cultural background of the participants, and especially the characteristics that culturally affect group dynamics need to be studied further, as yet another challenge.

Another limitation is the gender-imbalance of the sample, since most participants were male. Although there were efforts to keep the participant sample gender-balanced, this became a very challenging task in the researcher's department, where the vast majority of the student population is male. However, previous studies show that there are significant behavioral differences between males and females in issues that are relevant to our work, like competition [56] risk-taking [57], cooperation [58], altruism [59,60], honesty [61], etc. This implies that the present results should be seen under the light of female under-representation in the sample.

From the analysis of individual feelings during working processes, it should be noted that for successful cooperation of the team members and possible increases in the efficiency of the team, members could adopt roles that match their preferences and personalities. For example, non- leader types seem to feel embarrassed to express opinions and a cooperative task could give them the chance to contribute in more implicit/introverted manners. In addition, managers, task designers and others engaged in work organizational issues, need to remember that certain types of people feel that their opinions are not easily accepted or heard by the team. Perhaps different ways of expressing members' input could be considered that allow non-leader types to express their opinions without feeling intimidated. Finally, future work assisting computer systems could further explore personalization issues and adapt to their users preferences by requiring different types of contributions from the different people.



To conclude, the present work, apart from providing important research findings regarding intra-group collaboration, also reveals different challenges in the study of this complex field. Factors like the nature of the task (type, duration, etc), the types of participants (personalities, cultural background, prior knowledge of each other, etc), the level of computer mediation (distant collaboration, face to face with computer support, etc) and the type of group (large, small, homogeneous, etc) seem to play an important role, affecting group processes.

**Funding:** This research received no external funding.

**Conflicts of Interest:** The authors declare no conflict of interest.

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
