# Peer review of "Compatibility of Small Team Personalities in Computer-Based Tasks"

_challenges, doi:10.3390/challe10010029_

Round 1

Reviewer 1 Report

The submitted paper provides a convincing rationale why the research topic is relevant and points at the knowledge gaps that exist in that area.

The experimental approach presented in the study is interesting and fits into the general trend in economics, psychology and management sciences.

The conclusions and the discussion of the results is insightful and sober. I especially appreciate the clear identification of the limitations of the study resulting from the artificial study environment (experiment/game – not real life situations), size of the studied group, national homogeneity of experiment participants, age and occupational homogeneity of participants.

The future directions of research are well defined.

In my opinion the paper should be accepted as it presents an interesting (pilot) study that opens many doors for further research.

For the sake of clarity, I suggest that the author explains if the research focus on computer-based tasks results from the fact that it reflects most of the real-time situations or from the assumption that different personality compositions are needed for the effective functioning of teams performing such tasks (in comparison to non-computer-based facts).

More could be said about virtual vs. face-to-face team work in the context of computer-based tasks.

Author Response

Response to Reviewer 1

First of all, I would like to thank the reviewer for the effort put into reviewing the manuscript. I am particularly grateful for the kind words and also for the clarification point raised. The reviewer raises 2 important issues:

Point 1: Whether the study reflects real life work situations where computer-based tasks are a part of the everyday work environment or whether the study builds on the assumption that different personality compositions are needed for the effective functioning of teams in computer-based tasks vs tasks that do not require the use of a computer.

Response 1: This is a very good point and both issues apply here. The following text was added in the introduction to clarify the issue, further:

“Finally, the work is structured in a way that it reflects real life work situations (e.g. free communication of participants, participants known to each other, familiar space, etc) but it also builds on the assumption that computer-based tasks differ qualitatively from tasks that do not require the use of a computer, since technology changes the very nature of work and does not simply function as yet another office tool [Hlynka, 1996].”

Point 2: The reviewer asks for more discussion regarding virtual vs. face to face team work in computer mediated tasks.

Response 2: As required by the reviewer, I have now added literature on the comparison of face-to-face and virtual teams for computer mediated tasks. The text added is:

“In addition, for computer-based tasks when virtual teams and face-to-face ones were compared, the complexity of the endeavor became apparent, since many factors seem to play an important role in the effectives on a team [Majcharzal et al 2000]. It seems that compared to face-to-face teams, virtual ones need good supporting tools in order to perform their tasks and to increase group cohesiveness [Ehsan et al, 2008]. Face-to-face teams seem to outperform virtual ones under certain circumstances. For example, face-to-face teams are better for decision making tasks and for tasks with a single right answer [O’Neill et al 2016], they also report higher levels of satisfaction, although their communication effectiveness seems to be similar to the virtual teams [Warkentin et al, 2007]. The complexity of the field requires further study.”   

In addition, following the suggestion by both reviewers, I have now split the references in the introduction to show which ones correspond to face to face interaction and which to computer-based communication:

“Previous works study team members’ compatibility in a variety of environments and tasks, like face-to-face interaction [Belbin 2010, Gilley et al 2010, Furumo et al 2008, Halfhill et al 2005, Mathieu et a; 2008, Muchinsky et al 1987], computer based communication [Furumo et al 2008, Holton 2001, Lykourentzou et al 2016], etc.”

Moreover, and again after the suggestion by both reviewers, I have also included numerous references to different studies focusing on face to face tasks:

“Due to its importance, there are numerous works that study intra-group cooperation and they identify different important factors for effective cooperation like group size [Isaac and Walker, 1988; Isaac, Walker & Williams, 1994; Barcelo & Capraro, 2015; Capraro & Barcelo, 2015; Pereda et al, 2019], moral preferences [Capraro & Rand, 2018] and group members’ cooperation motives [Capraro et al, 2014; Engel & Zhurakhovska, 2016]. There are fewer works that focus on individual group member’s personality aspects and the why these personalities are compatible or not with the other group members [Lykourentzou et al 2016].”  

Reviewer 2 Report

This paper reports an experiment exploring how personality balance affects team performance in a computer based task requiring cooperation. The paper is interesting, but has several limitations, that I think could be addressed in a revision.

Comments:

1) The language should be more precise. For example, the sentence “Previous works study team members’ compatibility in a variety of environments and tasks, like face-to-face interaction, computer based communication, etc. [1, 3, 6, 7, 8, 9, 10, 11]” is very imprecise, because it is not clear what the references refer to. For example, if a reference talk about face-to-face interaction, then the authors should put it next to “face-to-face interaction”

2) The authors totally neglect the huge literature on cooperation. They say: “To the author’s best knowledge however, there are limited works that examine team compatibility within short cooperative activities.” Well, there is a huge amount of literature studying group cooperation. For example, the size of the group matters (Isaac and Walker, 1988; Isaac, Walker & Williams, 1994; Barcelo & Capraro, 2015; Capraro & Barcelo, 2015; Pereda et al, 2019); incentives matter (Capraro et al, 2014; Engel & Zhurakhovska, 2016); moral preferences matter (Capraro & Rand, 2018). This literature should be mentioned.

3) The sample was very small. This is an important limitation, especially in light of the Replicability Crisis (Open Science Collaboration, 2015). This should be discussed.

4) Another limitation regards the gender-imbalance of the sample, made mostly by males. It is known that females and males behave differently in a number of situations, including competition (Niederle & Vesterlund, 2007), risk-taking (Byrnes et al, 1999), cooperation (Rand, 2017), altruism (Rand et al, 2016; Brañas-Garza et al, 2018), and honesty (Capraro, 2018). Whether this impacts the results should be discussed

5) I noticed several typos. Please revise the writing carefully

References

Barcelo H, Capraro V (2015) Group size effect on cooperation in one-shot social dilemmas. Scientific Reports 5, 7937.

Brañas-Garza P, Capraro V, Rascón-Ramírez E (2018) Gender differences in altruism on Mechanical Turk: Expectations and actual behavior. Economics Letters 170, 19-23.

Byrnes JP, Miller DC, Schafer WD (1999) Gender differences in risk taking: A meta-analysis. Psychological Bulletin, 125, 367-383.

Capraro V (2018) Gender differences in lying in sender-receiver games: A meta-analysis. Judgment and Decision Making 13, 345-355.

Capraro V, Barcelo H (2015) Group size effect on cooperation in one-shot social dilemmas II: Curvilinear effect. PLoS ONE 10, e0131419.

Capraro V, Jordan JJ, Rand DG (2014) Heuristics guide the implementation of social preferences in one-shot Prisoner’s Dilemma experiments. Scientific Reports 4, 6790.

Capraro V, Rand DG (2018) Do the right thing: Experimental evidence that preferences for moral behavior, rather than equity and efficiency per se, drive human prosociality. Judgment and Decision Making 13, 99-111.

Engel, C., & Zhurakhovska, L. (2016). When is the risk of cooperation worth taking? The prisoner’s dilemma as a game of multiple motives. Applied Economics Letters, 23(16), 1157-1161.

Isaac RM, Walker JM (1988) Group size effects in public goods provision: The voluntary contribution mechanisms. The Quarterly Journal of Economics, 103, 179-199

Isaac RM, Walker JM, William AW (1994) Group size and the voluntary provision of public goods. Journal of Public Economics, 54, 1-36.

Open Science Collaboration. (2015). Estimating the reproducibility of psychological science. Science349(6251), aac4716.

Niederle M, Vesterlund L (2007) Do women shy away from competition? Do men compete too much? The Quarterly Journal of Economics, 122, 1067-1101.

Pereda M, Capraro V, Sánchez A (2019) Group size effects and critical mass in public goods games. Scientific Reports 9, 5503.

Rand DG (2017) Social dilemma cooperation (unlike Dictator Game giving) is intuitive for men as well as women. Journal of Experimental Social Psychology, 73, 164-168.

Rand DG, Brescoll VL, Everett JAC, Capraro V, Barcelo H (2016) Social heuristics and social roles: Intuition favors altruism for women but not for men. Journal of Experimental Psychology: General 145, 389-396.

Author Response

Response to Reviewer 2

First of all, I would like to thank the reviewer for the effort put into reviewing the manuscript. The points raised will certainly improve the quality of the manuscript. Thus, all reviewer comments were very helpful and are addressed as follows:

Point 1. The language should be more precise. For example, the sentence “Previous works study team members’ compatibility in a variety of environments and tasks, like face-to-face interaction, computer based communication, etc. [1, 3, 6, 7, 8, 9, 10, 11]” is very imprecise, because it is not clear what the references refer to. For example, if a reference talk about face-to-face interaction, then the authors should put it next to “face-to-face interaction”.

Response 1: The sentence has changed to show exactly which references correspond to face-to-face interaction and which to computer-based communication. The text was altered as follows:

“Previous works study team members’ compatibility in a variety of environments and tasks, like face-to-face interaction [Belbin 2010, Gilley et al 2010, Furumo et al 2008, Halfhill et al 2005, Mathieu et a; 2008, Muchinsky et al 1987], computer based communication [Furumo et al 2008, Holton 2001, Lykourentzou et al 2016], etc.”

Point 2: The authors totally neglect the huge literature on cooperation. They say: “To the author’s best knowledge however, there are limited works that examine team compatibility within short cooperative activities.” Well, there is a huge amount of literature studying group cooperation. For example, the size of the group matters (Isaac and Walker, 1988; Isaac, Walker & Williams, 1994; Barcelo & Capraro, 2015; Capraro & Barcelo, 2015; Pereda et al, 2019); incentives matter (Capraro et al, 2014; Engel & Zhurakhovska, 2016); moral preferences matter (Capraro & Rand, 2018). This literature should be mentioned.

Response 2: The reviewer was right and the way the sentence was written implied limited research. I have now changed the relevant part to include the references provided by the reviewer and to focus on the issue of intra-group personality balance. The new text is:

“Due to its importance, there are numerous works that study intra-group cooperation and they identify different important factors for effective cooperation like group size [Isaac and Walker, 1988; Isaac, Walker & Williams, 1994; Barcelo & Capraro, 2015; Capraro & Barcelo, 2015; Pereda et al, 2019], moral preferences [Capraro & Rand, 2018] and group members’ cooperation motives [Capraro et al, 2014; Engel & Zhurakhovska, 2016]. There are fewer works that focus on individual group member’s personality aspects and the why these personalities are compatible or not with the other group members [Lykourentzou et al 2016].”    

Point 3: The sample was very small. This is an important limitation, especially in light of the Replicability Crisis (Open Science Collaboration, 2015). This should be discussed.

Response 3: I have now put more emphasis on the matter in the light of the Replicability Crisis, as suggested by the reviewer. The test added is:

“Although the present work is an observational study that does not imply causality (as also reflected in the statistical analysis used), it is important to note that the small sample size possibly affects the reproducablity of results [Open Science Collaboration, 2015]. Replication studies are needed before certain conclusions can be drawn during which the present work can function as a pilot study.”

Point 4: Another limitation regards the gender-imbalance of the sample, made mostly by males. It is known that females and males behave differently in a number of situations, including competition (Niederle & Vesterlund, 2007), risk-taking (Byrnes et al, 1999), cooperation (Rand, 2017), altruism (Rand et al, 2016; Brañas-Garza et al, 2018), and honesty (Capraro, 2018). Whether this impacts the results should be discussed.

Response 4: I have now added in the discussion section, the following text that stresses the importance of gender balanced sample, as suggested by the reviewer:

“Another limitation regards the gender-imbalance of the sample, since most participants were male. Although there were efforts to keep the participant sample gender-balanced, this became a very challenging task in the researcher’s department, where the vast majority of the student population are male. However, previous studies show that there are significant behavior differences between males and females in issues that are relevant to our work, like competition [Niederle & Vesterlund, 2007] risk-taking [Byrnes et al, 1999], cooperation [Rand, 2017], altruism [Rand et al, 2016; Brañas-Garza et al, 2018], honesty [Capraro, 2018], etc. This implies that the present results should be seen under the light of female under-representation in the sample.”

Point 5: I noticed several typos. Please revise the writing carefully.

Response 5: Typos have been corrected.